# Factors associated with breast lesions among women attending select teaching and referral health facilities in Kenya: A cross-sectional study

Josephine Nyabeta Rioki[ID][1]*, Marshal Mweu[ID][2], Emily Rogena[3], Elijah M. Songok[4], Joseph Mwangi[4], Lucy Muchiri[1]

1 Department of Human Pathology, Faculty of Health Sciences, University of Nairobi, Nairobi, Kenya, 2 Department of Public and Global Health, Faculty of Health Sciences, University of Nairobi, Nairobi, Kenya, 3 Department of Human Pathology, School of Medicine, Jomo Kenyatta University of Agriculture and Technology, Nairobi, Kenya, 4 Kenya Medical Research Institute (KEMRI), Nairobi, Kenya

* jrioki@uonbi.ac.ke

## Abstract

Despite extensive research on the risk factors for breast cancer, little is known about the factors contributing to other breast lesions, of which some pose an increased risk for this disease. The objective of this study was to identify the factors associated with breast lesions among women presenting to referral facilities in Kenya for care between December 2016 and December 2019. An analytical cross-sectional study design was used to investigate the factors associated with breast lesions among 651 women with breast lumps. Data was collected using a semi-structured questionnaire. A multivariable logistic regression model was used to evaluate the impact of the factors on the breast lesions. The median age of participants was 30 years (range: 16−87), with the majority having secondary education and belonging to the Kikuyu ethnic group. Parity, exercise, and contraceptive use were significant factors identified. Nulliparous women had significantly lower odds of developing malignant [aOR: 0.11; 95% CI: 0.05–0.26] and suspicious [aOR: 0.23; 95% CI: 0.07–0.71] lesions. Regular exercise was associated with lower odds of both malignant and suspicious lesions. Conversely, contraceptive use increased the odds of developing atypical [aOR: 0.92; 95% CI: 0.28–2.98] lesions, suspicious [aOR: 0.33; 95% CI: 0.14–0.78], and malignant lesions [aOR: 0.31; 95% CI: 0.18–0.55]. Exercise, parity, and hormonal contraception were independently found to be significant factors associated with breast lesions in this study. These findings underscore the necessity for personalized risk reduction strategies and emphasize the importance of understanding the risk factors for both benign and malignant breast disease to inform public health policies.

**Data availability statement:** The data underlying the results presented in the study are available in Harvard Dataverse. https://doi.org/10.7910/DVN/2OXWTA.

**Funding:** This research was supported by a grant from National Research Fund, Kenya (Postgraduate grant 2016) to JNR. The funder did not participate in the conduct of the research. The funders had no role in study design, data collection and analysis, decision to publish, or preparation of the manuscript.

**Competing interests:** The authors have declared that no competing interests exist.

## Introduction

Breast lesions are a significant concern in women's health, with varying degrees of benign and malignant conditions affecting women globally. In Kenya, the prevalence of breast lesions remains a public health challenge, necessitating a deeper understanding of associated factors. The most frequently diagnosed cancer among women worldwide is breast cancer (BC), also referred to as a malignant breast lesion. About 80% of patients with the disease are individuals aged >50 [1]. The incidence of BC is higher in developed countries, but mortality rates are greater in developing countries [2,3]. Women of European descent have higher incidence rates, while women of African descent have lower incidence rates [4]. Kenya has one of the highest breast cancer mortality rates worldwide, with 80% of women diagnosed unlikely to live beyond five years [5]. Alarmingly, this challenge is expected to worsen, as the incidence of breast cancer in Kenya is projected to rise by a staggering 35% by 2025 [6].

Approximately 50% of breast cancer cases can be attributed to well-documented risk factors, such as increasing age (with the majority of cases occurring in women over 50), sex, family history of breast cancer (e.g., BRCA1 or BRCA2 gene mutations), and lifestyle factors like alcohol consumption or obesity. However, the remaining cases lack a clear explanation, as they are not associated with any of these established factors [7]. Importantly, even among individuals with known risk factors, such as those carrying BRCA mutations, the likelihood of developing breast cancer varies significantly based on additional factors, including lifestyle, environmental exposures, and genetic modifiers. This highlights the complex interplay of genetic, environmental, and hormonal influences in breast cancer development.

Risk factors for BC are classified into modifiable and non-modifiable factors. Modifiable risk factors include education level, parity, breastfeeding, passive smoking, and obesity. Non-modifiable risk factors include age, age at menarche, menopause, family history of any type of cancer, and family history of BC [8,9].

Global studies have examined the impact of reproductive factors on BC risk, including age at menarche, age at first childbirth, age at menopause, parity, breastfeeding, number of pregnancies, and the occurrence of abortion [10,11]. Existing literature indicates that changes in reproductive patterns, such as lower parity, delayed pregnancies, and shorter breastfeeding duration, contribute to an increased susceptibility to BC in women [11]. Research has also suggested that prolonged exposure to endogenous estrogen due to early menarche, delayed first childbirth, late menopause, or exogenous exposure through hormone replacement therapy or oral contraceptive use, are associated with a higher risk of BC [12]. Physical exercise has been shown to affect the fourteen hallmarks of breast cancer, thereby preventing the development of the disease [13].

Though extensive research has been done focusing on the factors associated with BC, little is documented on the factors associated with other breast lesions such as benign, atypical, and suspicious lesions. Factors associated with benign breast lesions also warrant attention. Benign breast diseases, including fibroadenomas, fibrocystic changes, and mastitis, are influenced by hormonal, genetic, and lifestyle factors. Hormonal imbalances, such as elevated estrogen levels during the

menstrual cycle, have been implicated in the development of fibrocystic changes [14]. Similarly, genetic predispositions, including polymorphisms in estrogen receptor genes, may increase susceptibility to benign breast conditions [15]. While these conditions are generally non-malignant, they may pose a higher risk of malignant transformation in certain cases, underscoring the need for regular monitoring and early intervention. Some of these lesions are associated with the risk of developing BC. Notably, among women diagnosed with invasive breast cancer, it is reported that 30% had benign breast disease previously [16]. It is further reported that among all atypical breast lesions, atypical ductal hyperplasia (ADH) has the highest likelihood of progressing to malignancy [17].

Since there exist differences in population structure, cultural factors, and the adoption of sedentary lifestyles across different regions in Kenya, understanding these contextual factors is crucial to inform the development of personalized risk reduction strategies and guide relevant public health policy. Therefore, the objective of this study was to identify the factors associated with breast lesions among women presenting to referral facilities in Kenya for care.

## Materials and methods

### Ethical considerations

Approval to conduct this study was granted by the Kenyatta National Hospital and the University of Nairobi Ethics and Research Committee [study no. P334/04/2016] as well as the institutional review boards of the select facilities. All participants provided written informed consent. The study was conducted following the Helsinki Declaration.

### Study design

This was a cross-sectional study aimed at investigating the factors associated with breast lesions (benign, atypical, suspicious, and malignant breast lesions) among women attending select teaching and referral health facilities in Kenya between December 2016 and December 2019. The study was reported as per the STROBE guidelines for reporting observational studies [18]. This work is part of a larger study, [reference number P334/04/2016] whose main objective was to characterize breast cancer phenotypes and genotypes among women diagnosed with breast cancer at selected teaching and referral health facilities in Kenya.

### Study sites

This study was conducted in two referral health facilities in Kenya (Kenyatta National Hospital [KNH] and Nakuru County Referral Hospital [NCRH]). These sites were selected because they provide specialized curative and diagnostic services. The two facilities are among the referral hospitals in Kenya where complex cases (breast cancer as one of such) are managed.

### Study population, eligibility, and selection of participants

The study population consisted of all women with palpable breast mass presenting to the fine needle aspirate and surgical clinics at NCRH and KNH for care. The patients were referred from other health facilities for evaluation of their breast mass and further management in the referral facilities. To participate in the study, women were required to have a palpable breast mass on physical examination, a history of spontaneous growth, and willing to give written informed consent. Women with a history of breast mass that resulted from physical trauma and those who declined to give written consent were excluded from the study.

### Outcome definitions

A breast lesion was a mass classified as benign, atypical, suspicious of malignancy, or malignant (cancerous) based on cytology as defined by the International Academy of Cytology Yokohama System (IAC YS) [19,20]. Notably, the IAC YS is

a standard tool used by cytopathologists and pathologists within the country to assure standardized diagnoses and thus minimize inter-observer variability.

## Data collection and study variables

Three research assistants were recruited and trained to assist with data collection. Quantitative data were collected using structured questionnaires. Information gathered included: demographics (age, ethnicity, education level and employment status), family history of breast cancer, reproductive factors (menstrual cycle, menstrual status, parity), hormonal contraceptive use, and lifestyle factors (history of smoking, alcohol history, and exercise). The factors assessed in this study are summarised in Table 1.

## Data processing and statistical analysis

Responses for qualitative variables from the questionnaires were coded before data entry. The data were entered in an MS Excel spreadsheet and exported to Stata v13 software for cleaning and analysis. The Stata code for this analysis is available as supporting information [21]. Continuous variables were summarized using medians and ranges. For categorical variables, frequencies and percentages were computed. To assess the association between the study independent variables and the breast lesion outcome, a univariable multinomial logistic regression model was used. At this stage, significance was considered at $P \leq 0.2$. Due to the small number of observations in some categories of contraceptive use, this variable was recategorized into a binary variable: user versus non-user.

Qualifying variables from the univariable analysis were offered to a multivariable multinomial logistic regression model and their significance evaluated at a strict $P<0.05$. To minimize confounding, non-significant variables were only excluded from the model when the coefficients of the remaining variables did not change by more than 30% [22]. The validity of the final parsimonious model was assessed by testing the assumption of independence irrelevant alternatives using the Hausman-McFadden test [23]. The null hypothesis for the test is that the model is valid at $P>0.05$.

**Table 1. Independent variables and their measurements.**

| Variables | Measurements |
|---|---|
| Age (continuous) | Captured in years |
| Level of education (ordinal) | Assessed in four levels: None, primary, secondary, and tertiary |
| Ethnicity (nominal) | This captures the major tribes in Kenya: Kikuyu, Luo, Luhya, Kalenjin, Kamba and minor tribes indicated as others |
| Employment status (nominal) | Captured as employed or unemployed |
| Menstrual cycle (nominal) | Captured as regular or irregular |
| Menstrual status (nominal) | Captured as premenopausal or post-menopausal |
| Parity (nominal) | Captured as nulliparous or parous |
| Contraception use (nominal) | Captured as those who had ever used either implant, oral and injection forms at any given time, or implant and oral only, injection only, oral only, implant only and never used any method |
| Family history of BC (nominal) | Captured as yes for those with family history or no for those without family history |
| Smoking history (nominal) | Captured as yes for those who have ever smoked or no for those who have never smoked. |
| Alcohol history (nominal) | Captured as yes for those who have ever taken alcohol and no for those who have never taken alcohol |
| Exercises (nominal) | Captured as yes for those who perform regular exercises and no for those that do not |

## Results

### Descriptive statistics

A total of 651 women with breast lumps were eligible for inclusion in the study. The median age for the participants was 30 years (Range: 16–97 years). Most women had attained secondary education (45.9%, n = 299). Kikuyus were the majority (54.5%, n = 355) with breast lumps. Most participants were employed (54.5%, n = 355). The majority (83.3%, n = 542) of the participants were premenopausal. Among the study participants, 10.5% (n = 68) were diagnosed with malignant lesions (breast cancer), 4.2% (n = 27) with lesions suspicious of malignancy, 1.8% (n = 12) had atypical cytologic findings, and 83.6% (n = 544) had benign lesions. The descriptive statistics for the factors associated with breast lesions are displayed in Table 2.

### Logistic regression analyses

**Univariable analysis.** Table 3 shows the results from the univariable analysis. Notably, age, ethnicity, level of education, employment status, menstruation status, parity, contraceptive use, family history of BC and exercise were significant at this stage ($P \leq 0.20$) and thus eligible for inclusion in the multivariable analysis.

### Multivariable analysis

Based on results from the multivariable analysis, parity, contraceptive use, and exercises were significantly associated with breast lesions at 5% significance level (Table 4).

The odds of atypical, suspicious, and malignant diagnoses in nulliparous women were about four-fifths, a quarter, and one-tenth, respectively, that of parous women [aOR: 0.11; 95% CI: 0.05–0.26; aOR: 0.23; 95% CI: 0.07–0.71; aOR: 0.79; 95% CI: 0.21–2.9] controlling for exercises and contraceptive use.

The odds of atypical, suspicious and malignant diagnoses among exercising women were roughly nine-tenths [aOR: 0.92; 95% CI: 0.28–2.98], one-third [aOR: 0.33; 95% CI: 0.14–0.78]and one-third [aOR: 0.31; 95% CI: 0.18–0.55], respectively, that of non-exercising women, controlling for parity and contraceptive use.

The odds of diagnosing atypical, suspicious and malignant lesions in women who use contraceptives were more than five and half times [aOR: 5.73; 95% CI: 1.14–28.94], about six times [aOR: 6.02: 95% CI: 1.73–20.95] and two times, [aOR: 1.82; 95% CI: 0.99–3.34], respectively, controlling for parity and exercises.

## Discussion

The key findings of this study suggest that parity, exercise, and contraceptive use are significant factors associated with breast lesions among women in Kenya. Nulliparity was found to have a substantial protective effect against the development of both suspicious and malignant breast lesions. This is a contradictory finding to other studies in the literature that identified parity as protective in the development of malignant breast lesions, especially those that are ER+/PR+ [24,25]. However, our findings correlate with some studies that have reported parous women to be more likely to develop triple-negative breast cancer [26,27]. This means that parity is not protective against all types of malignant breast lesions as conventionally documented. In fact, Nichols et al. [28] observed that recent childbirth slightly increased a woman's risk of breast malignancy – the risk peaking five years postpartum, but declining thereafter. Although nulliparous women had lower odds of developing atypical lesions, the difference was insignificant. Nevertheless, there is an indication that the likelihood of developing atypical breast lesions is lower in women who have never given birth.

In this study, physical exercise was significantly associated with a lower likelihood of malignant and non-malignant breast lesions. Women who reported engaging in exercise had lower odds of developing these lesions compared to those who did not exercise [aOR = 0.33, 95% CI: 0.14, 0.78 for suspicious lesions; aOR = 0.31, 95% CI: 0.18, 0.55 for malignant lesions; p < 0.001]. There is strong evidence to support that physical exercise has a significant impact on reducing the odds of developing malignant

**Table 2. Descriptive statistics of factors associated with breast lesions among women with breast lumps attending two select teaching and referral hospitals in Kenya (n = 651).**

| Variable | Category | Frequency n (%) | Median (Range) |
|---|---|---|---|
| Age | – | – | 30 (16–97) |
| Level of education | None | 4(0.6) | |
| | Primary | 167(25.7) | |
| | Secondary | 299(45.9) | |
| | Tertiary | 181(27.8) | |
| Ethnicity | Kamba | 48(7.4) | |
| | Kalenjin | 26(4) | |
| | Luhya | 32(4.9) | |
| | Luo | 64(9.8) | |
| | Kikuyu | 355(54.5) | |
| | Others | 126(19.4) | |
| Employment status | Yes | 355(54.5) | |
| | No | 296(45.5) | |
| Menstrual cycle | Regular | 569(87) | |
| | Irregular | 82(12) | |
| Menstrual status | Pre-menopause | 542(83.3) | |
| | Menopause | 109(16.7) | |
| Parity | Parous | 344 (52.8) | |
| | Nulliparous | 307(47.2) | |
| Contraception use | Injection+Implant only | 4(0.6) | |
| | Oral+Injection+Implant | 6(0.9) | |
| | Oral+Implant only | 13(2) | |
| | Oral+Injection only | 19(2.9) | |
| | Implant only | 26(4) | |
| | Injection only | 65(10) | |
| | Oral only | 190(29.3) | |
| | None | 326(50.2) | |
| Family history of BC | Yes | 32(4.9) | |
| | No | 618 (95.1) | |
| Smoking history | Yes | 12(1.8) | |
| | No | 639(98.2) | |
| Alcohol history | Yes | 40 (6.1) | |
| | No | 611(93.9) | |
| Exercise | Yes | 326(50.1) | |
| | No | 325(49.9) | |

breast lesions [29,30]. This is mainly because exercise modulates BMI and reduces obesity, both risk factors for malignant lesions of the breast [31]. Physical activity is known to modulate hormonal and inflammatory pathways, which could explain its protective effect against malignant breast lesions [32]. In addition, physical activity is associated with a significantly delayed onset of BC among breast cancer gene 1 and breast cancer gene 2 (BRCA1/2) mutation carriers [32,33]. This is because exercise enhances the body's Deoxyribonucleic acid (DNA) repair mechanisms, potentially correcting mutations that could lead to cancer [34]. Though the odds of developing atypical breast lesions in this population were lower, the difference between those who reported having physical exercise and those who had never had physical exercise was not statistically different.

**Table 3. Univariable analysis of factors associated with breast lesions among women attending two select referral hospitals in Kenya.**

| Factors | Atypical | Suspicious | Malignant | P-value |
|---|---|---|---|---|
| | cOR (95% C.I) | cOR (95% C.I) | cOR (95% C.I) | |
| **Age in years***  | 1.02(0.98, 1.06) | 1.04(1.01, 1.06) | 1.06(1.04, 1.07) | <0.001 |
| **Ethnicity*** | | | | 0.149 |
| Kamba | 1.62(0.14, 18.51) | 0.46(0.06, 3.92) | 3.50(1.49, 8.21) | |
| Kalenjin | 2.36(0.21, 27.22) | 0.68(0.08, 5.77) | 0.73(0.15, 3.45) | |
| Luhya | 2.00(0.17,22.91) | 0.57(0.07, 4.85) | 1.23(0.37, 4.09) | |
| Luo | 1.79(0.25, 13.07) | $5.39 \times 10^{-6}$(-, -) | 0.55(0.17, 1.77) | |
| Kikuyu | 0.86(0.16, 4.50) | 0.84(0.34, 2.07) | 0.82(0.41,1.63) | |
| Others (Ref) | 1 | 1 | 1 | |
| **Level of education*** | | | | 0.008 |
| None | 0.00(0,-) | 0.00(0,-) | 6.22(0.59,65.92) | |
| Primary | 6.72(0.78, 58.17) | 6.27(1.76, 22.29) | 3.43(1.54, 7.68) | |
| Secondary | 4.06(0.48, 34.03) | 2.26(0.61,8.33) | 2.63(1.23, 5.62) | |
| Tertiary (Ref) | 1 | 1 | 1 | |
| **Employment status*** | | | | <0.001 |
| Employed | 0.97(0.31, 3.05) | 2.31(0.99, 5.36) | 3.75(2.03, 6.90) | |
| Unemployed (Ref) | 1 | 1 | 1 | |
| **Menstrual cycle** | | | | 0.335 |
| Irregular | 0.64(0.08, 5.01) | 0.27(0.04, 2.02) | 1.5(0.77, 2.94) | |
| Regular | 1 | 1 | 1 | |
| **Menstrual status*** | | | | <0.001 |
| Menopause | 2.19(0.58, 8.26) | 2.29(0.94, 5.62) | 4.32(2.50, 7.45) | |
| Pre-menopause (Ref) | 1 | 1 | 1 | |
| **Parity*** | | | | <0.001 |
| Nulliparous | 0.43(0.13, 1.45) | 0.15(0.05, 0.44) | 0.10(0.04, 0.22) | |
| Parous (Ref) | 1 | 1 | 1 | |
| **Contraception use*** | | | | <0.001 |
| **Yes** <br> **No (Ref)** | 6.34(1.38, 29.20) <br> 1 | 10.14(3.02, 34.08) <br> 1 | 3.52(2.00, 6.19) <br> 1 | |
| **Family history of BC*** | | | | 0.069 |
| Yes | 4.52(0.94, 21.84) | $2.35 \times 10^{-6}$(-, -) | 2.60(1.07, 6.30) | |
| No (Ref) | 1 | 1 | 1 | |
| **Smoking history** | | | | 0.532 |
| Yes | $3.67 \times 10^{-6}$(-,-) | $3.67 \times 10^{-6}$(-, -) | 2.74(0.72, 10.37) | |
| No (Ref) | 1 | 1 | 1 | |
| **Alcohol history** | | | | 0.953 |
| Yes | $1.18 \times 10^{-6}$(-, -) | 0.56(0.07, 4.24) | 0.91(0.31, 2.64) | |
| No (Ref) | 1 | 1 | 1 | |
| **Exercises*** | | | | <0.001 |
| Yes | 0.86(0.27, 2.71) | 0.36(0.16, 0.84) | 0.36(0.21, 0.62) | |
| No (Ref) | 1 | 1 | 1 | |

Reference category for outcome is benign.

*Variables eligible for inclusion in the multivariable model (*P*≤0.20), cOR: crude odds ratio, CI: confidence interval.

**Table 4. Multivariable analysis of factors associated with breast lesions among women attending two select teaching and referral hospitals in Kenya.**

| Factors | Values | Atypical | Suspicious | Malignant | P-value |
|---|---|---|---|---|---|
| | | aOR (95% C.I) | aOR (95% C.I) | aOR (95% C.I) | |
| Parity | Nulliparous | 0.79(0.21, 2.9) | 0.23(0.07, 0.71) | 0.11(0.05, 0.26) | <0.001 |
| | Parous | 1 | 1 | 1 | |
| Exercises | Yes | 0.92(0.28, 2.98) | 0.33(0.14, 0.78) | 0.31(0.18, 0.55) | <0.001 |
| | No | 1 | 1 | 1 | |
| Contraception | Yes | 5.73(1.14, 28.94) | 6.02(1.73, 20.95) | 1.82(0.99, 3.34) | 0.002 |
| | No | 1 | 1 | 1 | |

aOR: adjusted odds ratio

Our study established that participants who used hormonal contraceptives had higher odds of being diagnosed with malignant breast lesions. These findings compare with those of a study done in Jordan, which indicated that regular use of oral contraception [OCs] increased the risk of malignant lesions of the breast [35]. Long-term use of oral contraceptives has been shown to elevate the risk of breast malignancy in older (>55 years) women [36]. Contraception use has also been associated with a substantially increased risk of developing atypical and suspicious lesions in this study.

Concerning contraceptive usage, almost an equal number of participants had used compared to those who had not. Some studies have reported a no-method preference as the second most preferred [37]. Our findings compare with data from 47 countries, which indicated that 40.9% of women in need of contraception were not using any methods. Moreira and others have explored reasons for the non-use of contraception despite the unmet need. Major reasons for non-use included, health concerns, infrequent sex, and opposition from others [38]. Religious and cultural factors may be the other reasons for non-use in our study population.

In terms of the breakdown of various contraceptives and their preference among women, the oral contraceptive was preferred. This finding is consistent with those reported in the literature [39]. Our study also indicates multiple uses of hormonal contraception over time, where the same individuals had used oral, injection, or implant. Similar findings have been reported previously in the US [40]. Simultaneous use of more than one contraception during sex has also been reported in the literature [41].

Patients' demographic and socioeconomic characteristics are important in diagnosis of malignant breast lesions and management. Most participants in this study had a minimum of secondary education suggesting a level of breast health awareness and positive health-seeking behavior in this population. Concerning ethnicity, most participants were of the Kikuyu ethnic community. This is likely, as this is the second most populous ethnic group in Nakuru County and the largest ethnic group in Kenya. In addition, KNH is close to Central Kenya, a geographical region historically occupied by Kikuyus.

Overall, from the univariable analysis, the factors that were found to be significantly associated with breast lesions included age, ethnicity, level of education, employment status, family history of BC, and exercise. The findings from this study correlate with those of other studies. The risk of malignant lesions increases with age [42]. Several studies have reported racial and ethnic variation in BC where a minority of women present with higher-stage BC than women of European ancestry [42,43]. Such differences in race and ethnicity have also been discussed by others [42,44,45].

This study significantly associated education level and employment status with malignant lesions, consistent with other studies [46,47]. This is likely because employment and education are linked to increased awareness, quality of life, and exposure to other risk factors such as alcohol and smoking [48]. Family history on the other hand is an important risk factor for malignant lesions as demonstrated in our study. This important association has been demonstrated previously in other studies [49,50].

The findings of this study such as contraceptive use, age, menopausal status, exercise, and socioeconomic determinants, align with global trends reported in breast cancer research. However, certain region-specific determinants merit further attention. Unique risk factors specific to the Kenyan population may include cultural, environmental, and healthcare access. Cultural differences in dietary patterns, such as the high consumption of maize-based diets low in essential nutrients, may also play a role. While genetic testing is limited in Kenya, familial history of breast cancer remains a relevant risk factor. This warrants specific investigations within the Kenyan population to understand the risk in this setting.

## Strengths and limitations of the study

The study addresses a pressing public health issue in Kenya, where breast lesions are becoming increasingly prevalent. Conducting the study in teaching and referral hospitals ensures access to a population likely to represent a spectrum of cases, from benign lesions to potentially malignant ones. This setting also allows for better diagnostic accuracy and availability of medical records.

The research examines a comprehensive range of both modifiable (e.g., lifestyle factors like smoking, obesity, and alcohol use) and non-modifiable (e.g., family history, age at menarche) risk factors. This broad approach provides a more complete picture of the determinants of breast lesions, enhancing its utility for public health interventions.

With the rising burden of breast cancer in Africa and the increasing recognition of disparities in outcomes for women of African descent, this research is timely. It aligns with global efforts to reduce breast cancer mortality through early identification of risk factors and preventive interventions.

This study is not without limitations. Even though capturing variables polytomously (rather than dichotomously) affords more statistical information and allows for assessment of dose-response relationships, this was not possible for some set of variables. In particular, smoking and alcohol consumption are practices culturally frowned upon among women in many African rural populations, and as such, it is typical to see studies dichotomizing these variables. Moreover, as earlier pointed out, other variables were recategorized owing to the small number of observations in some categories. Assessment of the receptor status (estrogen and progesterone receptors and human epidermal growth factor receptor 2) could potentially yield additional vital statistical information. Nonetheless, conventionally in our facilities, patients are only screened for the receptor status once they have been diagnosed with breast cancer (malignancy) to guide effective therapy. Consequently, this status would only be expected for those women diagnosed with malignant breast lesions. Even so, considering the observational design of this study, we could not retrieve this receptor status for a majority of the patients with malignant lesions.

It was not possible to tell if the associations observed were caused by identified risk factors because of the study design employed. A cross-sectional study design limits causal inference. Conducting the study in select teaching and referral hospitals may limit the representativeness of findings to other populations or rural settings in Kenya. However, it was presumed that diagnosis is done in referral centers, and a spectrum of breast lesions could be identified with improved diagnostic accuracy. Selection bias is possible since there are those women who may not have accessed referral centers because they come from remote parts of Kenya. Self-reporting of the risk factors or medical history could have also introduced recall bias. The authors suggest that while the findings are valuable for generating hypotheses, future studies employing stronger designs, such as case-control or cohort studies, are necessary to validate the results. Enhanced methodologies, including objective exposure assessments and broader population sampling, are recommended to address the limitations and provide a clearer understanding of breast lesion risk factors in Kenya.

## Conclusions

From this study, exercise, parity, and hormonal contraception were found to be significant factors associated with benign and malignant breast lesions. These findings warrant consideration of breast disease risk reduction through empowering women to make informed reproductive choices and the choice of contraception use, and weight management.

## Supporting information

**S1 File. Harvard Dataverse: Replication data for: Factors associated with breast lesions among women attending select teaching and referral health facilities in Kenya: a cross-sectional study,** https://doi.org/10.7910/DVN/2OXWTA.
(DOCX)

## Acknowledgments

The authors wish to express their sincere gratitude to the study participants for contributing to the research. They also acknowledge the University of Nairobi′s Building Capacity for Writing Scientific Manuscripts (UANDISHI) Program at the Faculty of Health Sciences for the training on manuscript writing.

## Author contributions

**Conceptualization:** Josephine Nyabeta Rioki, Emily Rogena.

**Data curation:** Josephine Nyabeta Rioki, Marshal Mweu.

**Formal analysis:** Josephine Nyabeta Rioki, Marshal Mweu.

**Funding acquisition:** Josephine Nyabeta Rioki, Elijah M. Songok.

**Investigation:** Josephine Nyabeta Rioki, Marshal Mweu, Emily Rogena, Elijah M. Songok, Lucy Muchiri.

**Methodology:** Josephine Nyabeta Rioki, Marshal Mweu, Emily Rogena.

**Project administration:** Josephine Nyabeta Rioki, Emily Rogena, Lucy Muchiri.

**Resources:** Josephine Nyabeta Rioki, Elijah M. Songok.

**Supervision:** Marshal Mweu, Emily Rogena, Elijah M. Songok, Lucy Muchiri.

**Writing – original draft:** Josephine Nyabeta Rioki, Joseph Mwangi.

**Writing – review & editing:** Josephine Nyabeta Rioki, Marshal Mweu, Emily Rogena, Elijah M. Songok, Joseph Mwangi, Lucy Muchiri.

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
