## [Decision Letter · Decision Letter 0]

4 Oct 2024

Dear Dr. Rioki,

Thank you for submitting your manuscript to PLOS ONE. After careful consideration, we feel that it has merit but does not fully meet PLOS ONE’s publication criteria as it currently stands. Therefore, we invite you to submit a revised version of the manuscript that addresses the points raised during the review process.

We look forward to receiving your revised manuscript.

Kind regards,

Jie Yang, M.D.

Guest Editor

PLOS ONE

Journal Requirements:

“This research was supported by a grant from National Research Fund, Kenya (Postgraduate grant 2016) to JNR.

The funder did not participate in the conduct of the research.”

Reviewers' comments:

Reviewer's Responses to Questions

**Comments to the Author**

1. Is the manuscript technically sound, and do the data support the conclusions?

Reviewer #1: No

Reviewer #2: Partly

Reviewer #3: Partly

2. Has the statistical analysis been performed appropriately and rigorously?

Reviewer #1: No

Reviewer #2: Yes

Reviewer #3: Yes

3. Have the authors made all data underlying the findings in their manuscript fully available?

Reviewer #1: Yes

Reviewer #2: Yes

Reviewer #3: Yes

4. Is the manuscript presented in an intelligible fashion and written in standard English?

Reviewer #1: Yes

Reviewer #2: Yes

Reviewer #3: No

Reviewer #1: This case series study of 651 breast lump cases from two hospitals in Kenya. It may have its own value that it is a epidemiologic study from a country with limited resources. However, the association between lifestyle risk factors with breast lesions was studied in many previous studies, and the number of cases isn’t sufficient to support the study power in multinomial logistic regression.

Cases included several diagnosis and the authors categorized the cases in four groups for statistical analysis. Therefore, although the authors underlined they included several types of abnormal cases of the breast, the number of cases was not enough to verify the association of the risk factors and the diseases. I would recommend rather to divide the cases to two groups to non-benign cases and benign cases in a study of 651 cases in total.

In the body of the manuscript, the description are in detail in some parts but some important description needs be added or improved. For example, I think description of the rationale of the hypothesis of this study needs to be strengthened, and introduction of the medical system in Kenya needs explanation. For example, not many people outside Kenya are familiar with ‘Level 6’. In the methods, some parts - such as inclusion criteria and exclusion criteria - needs to be described in detail. In discussion, a detail discussion regarding study results could be summarized, and strength and limitation of the study needs to be added. In describing the results, the association found in this study could not be suggested as predictors of the outcomes since the study design was retrospective. In presenting odds ratios and confidence intervals in the tables, presenting number of cases and percentage in each cells together would help the authors to understand the main results. I’d prefer to omit Table 1 or send it to supplements.

Reviewer #2: Thanks for the chance to review your work. This is a cross-sectional study on breast lesions among women attending select teaching and referral health facilities in Kenya. I am concious that this study is a clinical paper based on clinical data gained from real-world practice.

1. Thus, it is important to reach a reliable conclusion depending on reasonable statistical analysis. I recommend that a statistic expert should conprehensively evaluate the methodology aspects.

2. Please clarify the inclusion and exclusion criteria.

3. Please consider to further discuss the potential risk factors for breast diseases in Kenya. What are the common ones that are shared among the world? What are the special ones limitted to Kenya?

4. The participants in this study were women attending select teaching and referral health facilities. Why you chose this kind of sample? Were they representative enough?

Reviewer #3: The manuscript presents an interesting topic regarding the factors associated with breast lesions. However, the writing throughout the manuscript needs to be rechecked and improved.

Specific Comments:

Major Comments:

-Improve the English writing.

-Re-check and reformat the references.

-The discussion needs to be rewritten; it is almost entirely related to breast cancer. The author mentions in the title, abstract, and introduction a focus on breast lesions rather than breast cancer.

Minor Comments:

Lines 27-29: This information should be in the methods section of the manuscript, not in the methods part of the abstract.

Line 47: The terms "transitioned" and "transitioning countries" are not particularly meaningful; consider using alternative words.

Lines 47-48: The information is not found in the cited reference. Additionally, avoid mentioning "black" or "white" people without specifying racial or ethnic categories or populations.

Line 48: Instead of “unlike,” consider using a different word.

Line 51: This statement is too general and requires more detail: “About half of breast cancers can be explained by known risk factors (such as age and female gender), while the other half may remain unknown [5]. However, even with known risk factors, it is not certain that females with other risk factors will develop breast cancer.” Additionally, when referring to females, it is unnecessary to include "female gender."

Lines 117-134: This section should be cited only; all information is already present in the references, or it should be paraphrased.

Line 203: The introduction of Table 6.3 appears abruptly.

Line 203: The statement "p < 0.2" is mentioned, yet the author refers to it as significant. This needs to be rechecked in the text.

**Do you want your identity to be public for this peer review?** For information about this choice, including consent withdrawal, please see our Privacy Policy

Reviewer #1: No

Reviewer #2: No

Reviewer #3: No

---

## [Author Response · Author response to Decision Letter 1]

24 Jan 2025

Dr. Josephine Nyabeta Rioki

Faculty of Health Sciences

Department of Human Pathology

University of Nairobi,

Kenya

24th January 2025.

Email: jrioki@uonbi.ac.ke

Dear reviewers,

PONE-D-24-31297: Factors associated with breast lesions among women attending select teaching and referral health facilities in Kenya: A cross-sectional study

We thank you for taking time to review our manuscript in consideration for publication in your Journal. We are grateful for your observations and suggestions. Your comments were addressed as follows:

Journal Requirements:

Comments 1:

Response:

PLOS ONE’S style requirements have been considered.

Comments 2:

Thank you for stating the following financial disclosure:“This research was supported by a grant from National Research Fund, Kenya (Postgraduate grant 2016) to JNR. The funder did not participate in the conduct of the research.” Please state what role the funders took in the study. If the funders had no role, please state: "The funders had no role in study design, data collection and analysis, decision to publish, or preparation of the manuscript."

Response:

Comments 3:

Your ethics statement should only appear in the Methods section of your manuscript. If your ethics statement is written in any section besides the Methods, please delete it from any other section.

Response:

Ethics statemnent has been deleted from the abstract and it is only appearing in materials and methods section

COMMENTS BY REVIEWER 1

Comment 1: This case series study of 651 breast lump cases from two hospitals in Kenya. It may have its own value that it is a epidemiologic study from a country with limited resources. However, the association between lifestyle risk factors with breast lesions was studied in many previous studies, and the number of cases isn’t sufficient to support the study power in multinomial logistic regression.

Response:

Even though the risk factors for breast lesions have been widely studied, there still remains a dearth of information on the factors that are associated with breast lesions among Sub-Saharan African women (Brinton et al. 2014). Understanding this set of factors may be key to tailoring interventions specific for this population. On sample size, Schwab (2002) contends that for a mutinomial logistic regression, a minimum of 10 cases per independent variable are necessary to sufficiently power it. With 12 predictors in our analysis, this translates to a sample of 120. Thus, our sample of 650 individuals adequately fulfils this requirement.

Comment 2: Cases included several diagnosis and the authors categorized the cases in four groups for statistical analysis. Therefore, although the authors underlined they included several types of abnormal cases of the breast, the number of cases was not enough to verify the association of the risk factors and the diseases.

I would recommend rather to divide the cases to two groups to non-benign cases and benign cases in a study of 651 cases in total.

Response:

Regrouping the outcome into a binary variable as suggested by the reviewer would result in loss of vital statistical information considering that the studied breast lesion categories hold distinct cytopathological definitions. Moreover, as argued in the previous comment, the sample of 651 cases sufficiently powers the multinomial logistic regression analysis performed in the study.

Comment 3: In the body of the manuscript, the description are in detail in some parts but some important description needs be added or improved. For example, I think description of the rationale of the hypothesis of this study needs to be strengthened, and introduction of the medical system in Kenya needs explanation. For example, not many people outside Kenya are familiar with ‘Level 6’.

Response: Improvement and addition of the description has been made in the manuscript (see page 4 and page 5 on the study sites )

In Kenya, we have referral and teaching health facilities/hospitals where specialized curative and diagnostics services are offered. Patients with breast masses for example are referred there from other health facilities in the country that do not have specialized human resource and equipment to render appropriate diagnoses.

Comment 4: In the methods, some parts - such as inclusion criteria and exclusion criteria - needs to be described in detail.

Response: Revision has been made (see page 6)

Comment 5: In discussion, a detail discussion regarding study results could be summarized, and strengths and limitation of the study need to be added.

Response: The discussion has been revised, and the strengths and study limitations have been included in the manuscript

Comment 6: In describing the results, the association found in this study could not be suggested as predictors of the outcomes since the study design was retrospective. In presenting odds ratios and confidence intervals in the tables, presenting number of cases and percentage in each cell together would help the authors to understand the main results.

Response:

The term ‘predictors’ has been changed to ‘factors’ throughout the manuscript as suggested. However, even though it would have been ideal to present numbers of cases and percentages for the cells in the univariable analysis, this is likely to crowd out the main results from the table. More so, in the multivariable table, since the odds ratios are already adjusted for confounding, adding numbers of cases and percentages would be statistically erroneous.

Comment 7:

I’d prefer to omit Table 1 or send it to supplements.

Response:

The STROBE guidelines for reporting observational studies (Von Elm et al. 2008) dictate that for cross-sectional studies ‘details of methods of assessment/measurement for each variable of interest’ should be captured in the methodology section. This promotes transparency and assures reproducibility of the methods. Therefore, the suggestion to transfer this information to the supplement section would undermine this guideline.

COMMENTS BY REVIEWER 2

Thanks for the chance to review your work. This is a cross-sectional study on breast lesions among women attending select teaching and referral health facilities in Kenya. I am conscious that this study is a clinical paper based on clinical data gained from real-world practice.

Comment 1: Thus, it is important to reach a reliable conclusion depending on reasonable statistical analysis. I recommend that a statistic expert should comprehensively evaluate the methodology aspects.

Response:

One of the co-authors (Dr. Mweu) is an epidemiologist/biostatistician and was involved in drafting/reviewing the methodological aspects of the study (particularly the statistical section) as well as the results portion of the manuscript. Consequently, some necessary revisions have been effected.

Comment 2: Please clarify the inclusion and exclusion criteria.

Response: Revision has been made to the inclusion and exclusion criteria (see page 6)

Comment 3: Please consider to further discuss the potential risk factors for breast diseases in Kenya. What are the common ones that are shared among the world? What are the special ones limited to Kenya?

Response:

An additional paragraph highlighting potential factors that could influence breast disease risk has been included (see page 16).

Comment 4: The participants in this study were women attending select teaching and referral health facilities. Why you chose this kind of sample? Were they representative enough?

Response: We chose women attending teaching and referral health facilities for this study because these institutions are typically well-equipped to handle diagnostic and treatment services for breast lesions. These facilities attract patients from diverse socio-demographic and geographic backgrounds, making them a strategic choice for capturing a wide range of cases.

While the sample was limited to women attending these facilities, the selection aimed to include a population reflective of those seeking medical care for breast-related issues in Kenya. However, we acknowledge potential limitations in generalizability, particularly to women who may not have access to healthcare services. This limitation is discussed in the manuscript under the "Study Limitations" section (page 17).

COMMENTS BY REVIEWER 3

The manuscript presents an interesting topic regarding the factors associated with breast lesions. However, the writing throughout the manuscript needs to be rechecked and improved.

Major Comments: Improve the English writing.

Response: English has been improved throughout the manuscript

Comment: Re-check and reformat the references.

Response: Reformatting of the references has been done as per PLOS ONE requirements

Comment:The discussion needs to be rewritten; it is almost entirely related to breast cancer. The author mentions in the title, abstract, and introduction a focus on breast lesions rather than breast cancer.

Response:

The discussion has been revised extensively to address this comment.

Comment: Lines 27-29: This information should be in the methods section of the manuscript, not in the methods part of the abstract.

Response: The information in Lines 27-29 has been deleted from the manuscript

Comment: Line 47: The terms "transitioned" and "transitioning countries" are not particularly meaningful; consider using alternative words.

Response: Paraphrasing of the paragraph has been done

Comment: Lines 47-48: The information is not found in the cited reference. Additionally, avoid mentioning "black" or "white" people without specifying racial or ethnic categories or populations.

Response: The description of the population has been revised as advised. The correct reference has been included

Comment: Line 48: Instead of “unlike,” consider using a different word.

Response: The whole sentence was revised.

Comment: Line 51: This statement is too general and requires more detail: “About half of breast cancers can be explained by known risk factors (such as age and female gender), while the other half may remain unknown [5]. However, even with known risk factors, it is not certain that females with other risk factors will develop breast cancer.” Additionally, when referring to females, it is unnecessary to include "female gender."

Response: Revision have been made

Comment: Lines 117-134: This section should be cited only; all information is already present in the references, or it should be paraphrased.

Response: Most of the content was deleted and citations made for the remaining information.

Comment: Line 203: The introduction of Table 6.3 appears abruptly.

Response: Relabeling of the table has been made (see page11)

Comment: Line 203: The statement "p < 0.2" is mentioned, yet the author refers to it as significant. This needs to be rechecked in the text.

Response:

Significance at the univariable analysis stage was gauged at a liberal P-value of ≤0.20 that aims to allow variables that are potentially negatively confounded have the opportunity to ‘express themselves’ in a multivariable analysis (Dohoo et al. 2012). Hence the term ‘significant’ here refers to the 20% cut-off level.

Thank you for the comments and suggestions by the reviewers, they have all been addressed as above.

I am looking forward to hearing from you.

References

Brinton, L.A., Figueroa, J.D., Awuah, B., Yarney, J., Wiafe, S, Wood, S, Ansong, D., Nyarko, K, Wiaffe-Addai, B., Clegg-Lamptey, J.N. 2014. Breast Cancer in Sub-Saharan Africa: Opportunities for Prevention. Breast Cancer Res. Treat., 144 (3): 467-478

Dohoo I., Martin, W., Stryhn, H. 2012. Methods in Epidemiologic Research, VER Inc. 1st Edition, Prince Edward Island, Canada.

Schwab, J.A. 2002. Multinomial Logistic Regression: Basic Relationships and Complete Problems. University of Texas, Austin, Texas

Von Elm E, Altman DG, Egger M, Pocock SJ, Gøtzsche PC, Vandenbroucke JP. The Strengthening the Reporting of Observational Studies in Epidemiology (STROBE) statement: guidelines for reporting observational studies. Journal of Clinical Epidemiology [Internet]. 2008 Apr [cited 2024 May 31];61(4):344–9.

Yours faithfully,

JNR

Dr. Josephine N. Rioki

---

## [Decision Letter · Decision Letter 1]

25 Feb 2025

Dear Dr. Rioki,

Thank you for submitting your manuscript to PLOS ONE. After careful consideration, we feel that it has merit but does not fully meet PLOS ONE’s publication criteria as it currently stands. Therefore, we invite you to submit a revised version of the manuscript that addresses the points raised during the review process.

We look forward to receiving your revised manuscript.

Kind regards,

Jie Yang, M.D.

Guest Editor

PLOS ONE

Journal Requirements:

Additional Editor Comments:

Thanks for submitting your revised work to PLOS ONE. Your manuscript has now been assessed by our editorial team and the previous peer experts. You will see that Reviewer #1 can not approve your paper this time and has further raised many serious problems. Please submit the point-by-point responses to Reviewer #1's comments. Additionally, Reviewer #3 has some minor suggestions as well. If you are prepared to undertake the work required, I would be pleased to reconsider my decision. Please note that this revision decision does not assure the acceptance of your work. If your revision work still can not meet the requirement of Reviewer #1, I may reject your manuscript or seek for the viewpoints of another reviewer to finally make an informed decision. Thanks for the opportunity to consider your work.

Reviewers' comments:

Reviewer's Responses to Questions

**Comments to the Author**

Reviewer #1: (No Response)

Reviewer #2: All comments have been addressed

Reviewer #3: All comments have been addressed

2. Is the manuscript technically sound, and do the data support the conclusions?

Reviewer #1: Partly

Reviewer #2: Yes

Reviewer #3: Yes

3. Has the statistical analysis been performed appropriately and rigorously?

Reviewer #1: No

Reviewer #2: Yes

Reviewer #3: Yes

4. Have the authors made all data underlying the findings in their manuscript fully available?

Reviewer #1: Yes

Reviewer #2: Yes

Reviewer #3: Yes

5. Is the manuscript presented in an intelligible fashion and written in standard English?

Reviewer #1: Yes

Reviewer #2: Yes

Reviewer #3: Yes

Reviewer #1: I believe the improvements in the background and discussion enhance the readers’ understanding of the study’s objectives and results more clearly. I appreciate the authors’ efforts in collecting data from a low-resource setting and sincerely hope to see the study develop further.

However, the current study has some limitations regarding the available variables. Specifically, dose information is not provided, and most variables are categorized as binary (yes or no). Additionally, important clinical characteristics, such as receptor status, are unavailable. As a result, the study’s implications are somewhat limited in contributing novel insights beyond what is already known about breast cancer and benign lesions.

The authors argue that the number of cases is sufficient for a multinomial logistic model. However, based on the large confidence intervals observed in several cells, it appears that some cells may have a small number of cases, which could lead to debatable conclusions.

Nevertheless, I hope the authors successfully find a suitable platform to share this valuable dataset.

Reviewer #2: Thanks for your response to my concerns, and I think my problems are addressed properly now.

Reviewer #3: The revised version addressed all previous comments and following are minor comments:

Line 325-326: need references

Line 363-364: need references

**Do you want your identity to be public for this peer review?** For information about this choice, including consent withdrawal, please see our Privacy Policy

Reviewer #1: No

Reviewer #2: No

Reviewer #3: No

---

## [Author Response · Author response to Decision Letter 2]

13 Mar 2025

Dr. Josephine Nyabeta Rioki

Faculty of Health Sciences

Department of Human Pathology

University of Nairobi,

Kenya.

12 March 2025.

Email: jrioki@uonbi.ac.ke

Dear Editor,

Thank you for the comments we received concerning our manuscript -PONE-D-24-31297R1: Factors associated with breast lesions among women attending select teaching and referral health facilities in Kenya: A cross-sectional study.

We appreciate the time taken to review our paper and we wish to thank the reviewers for their thoughtful feedback.

We have attached our response to the reviewers.

Looking forward for a favorable response.

Thank you.

JNR

Dr. Josephine Nyabeta Rioki

Dr. Josephine Nyabeta Rioki

Faculty of Health Sciences

Department of Human Pathology

University of Nairobi,

Kenya

12 March 2025.

Email: jrioki@uonbi.ac.ke

Dear reviewers,

PONE-D-24-31297R: Factors associated with breast lesions among women attending select teaching and referral health facilities in Kenya: A cross-sectional study

We thank you for taking time to review our manuscript in consideration for publication in the PLoS ONE Journal. We are grateful for your observations and suggestions. Your comments were addressed as follows:

COMMENTS BY REVIEWER 1

Comment 1: I believe the improvements in the background and discussion enhance the readers’ understanding of the study’s objectives and results more clearly. I appreciate the authors’ efforts in collecting data from a low-resource setting and sincerely hope to see the study develop further.

However, the current study has some limitations regarding the available variables. Specifically, dose information is not provided, and most variables are categorized as binary (yes or no).

Response:

On dose response and categorisation of variables, we firstly take note that this is a new comment raised by the reviewer (this had not been pointed out in the earlier feedback). Nonetheless, we agree with the reviewer’s assertion that capturing variables in a dose-response fashion would perhaps afford more useful statistical information than representing them binarily. However, for some variables e.g. smoking and alcohol histories, these practices are culturally frowned upon among women in many African rural populations and as such, it is typical to see studies dichotomising these variables. It is in this light that we recorded them as binary. Moreover, other few variables (as mentioned in the methods section) were recategorised owing to the small number of observations in some categories.

Comment 2: Additionally, important clinical characteristics, such as receptor status are unavailable. As a result, the study’s implications are somewhat limited in contributing novel insights beyond what is already known about breast cancer and benign lesions.

Response:

Again, we note that this is a new issue raised by the reviewer. We nevertheless appreciate that capturing and subsequently assessing this variable could potentially yield some vital statistical information. However, it is worthwhile pointing out that, conventionally in our facilities, patients are only screened for the receptor status (ER, PR and HER2) once they have been diagnosed with breast cancer (malignancy) so as to guide effective therapy. Consequently, this status would only be expected for those women diagnosed with malignant breast lesions. More so, considering the observational design of this study, we could not retrieve this receptor status for a majority of these malignant-lesion patients. Notwithstanding, we are persuaded that this study sheds light on factors associated with breast lesions amongst women in Sub-Saharan Africa – evidence that is glaringly scarce.

Comment 3: The authors argue that the number of cases is sufficient for a multinomial logistic model. However, based on the large confidence intervals observed in several cells, it appears that some cells may have a small number of cases, which could lead to debatable conclusions.

Response: As contended in our previous rebuttal letter, the sample available for this study was sufficiently sized to power the multinomial logistic regression. More importantly, the decision as to which which statistical model framework to adopt (multinomial vs binary logistic regression) rested on a delicate choice between (1) retaining universally accepted, standard cytopathological definitions of breast lesions (as per the International Academy of Cytology Yokohama System) and thus preserving critical statistical information inherent in the four breast lesion categories (multinomial logistic regression), and (2) upholding the statistical principle of parsimony (binary logistic regression being the simpler model). Considering that the patients in this study were independently classified into one of four distinctive breast lesion categories, with a view to retaining this useful information whilst keeping meaningful biological interpretability of the results, the multinomial logistic regression model framework was deemed an obvious choice. Albeit with a slight statistical shortcoming – wide confidence intervals for few variables. Despite this, statistically, the point estimates (odds ratios) should not be affected – thus the drawn conclusions should remain unchanged.

Comment 4: Nevertheless, I hope the authors successfully find a suitable platform to share this valuable dataset.

Response: The dataset for this study is already deposited in a public repository: https://doi.org/10.7910/DVN/2OXWTA

COMMENTS BY REVIEWER 2

The revised version addressed all previous comments and following are minor comments:

Line 325-326: need references

Line 363-364: need references

Response:

A reference has been added in line 238-239 that necessitated a reference (see page 14). Moreso,

a statement that required a reference has been expunged since it was a general statement (see page 15).

We thank you for your thoughtful feedback.

Yours faithfully,

JNR

Dr. Josephine N. Rioki

---

## [Decision Letter · Decision Letter 2]

1 Apr 2025

Dear Dr. Rioki,

Thank you for submitting your manuscript to PLOS ONE. After careful consideration, we feel that it has merit but does not fully meet PLOS ONE’s publication criteria as it currently stands. Therefore, we invite you to submit a revised version of the manuscript that addresses the points raised during the review process.

We look forward to receiving your revised manuscript.

Kind regards,

Jie Yang, M.D.

Guest Editor

PLOS ONE

Additional Editor Comments:

Dear authors, We have invited additional peer experts to assess your revised manuscript, and now the editorial process is completed. Regretably, 2 reviewers of 3 additional reviewers (Reviewer 4/5/6) recommended rejection and proposed many serious problems. Thus, based on all the feedbacks from reviewers, we regret to inform you that we can not consider your paper for publication in its current form. However, there are still 3 reviewers commending your paper. Thus, we have now decided that this major revision decision is the last chance to improve your manuscript and comprehensively respond to reviewers' comments. Please note that if your revision work can not meet the requirment of the reviewers, we will reject your paper without further inviting other experts. Considering the complicated circumstance of this paper, I will invite some internal editors to make the final editorial decision. Thanks for the chance to consider your work. If you have any questions, please do not hesitate to contact me or the editorial office. If reviewers want to discuss our editorial desicion, please contact me as well. Thank you very much.

Reviewers' comments:

Reviewer's Responses to Questions

**Comments to the Author**

Reviewer #3: All comments have been addressed

Reviewer #4: (No Response)

Reviewer #5: All comments have been addressed

Reviewer #6: All comments have been addressed

2. Is the manuscript technically sound, and do the data support the conclusions?

Reviewer #3: Yes

Reviewer #4: No

Reviewer #5: Partly

Reviewer #6: Yes

3. Has the statistical analysis been performed appropriately and rigorously?

Reviewer #3: Yes

Reviewer #4: No

Reviewer #5: Yes

Reviewer #6: Yes

4. Have the authors made all data underlying the findings in their manuscript fully available?

Reviewer #3: Yes

Reviewer #4: No

Reviewer #5: Yes

Reviewer #6: Yes

5. Is the manuscript presented in an intelligible fashion and written in standard English?

Reviewer #3: Yes

Reviewer #4: Yes

Reviewer #5: Yes

Reviewer #6: Yes

Reviewer #3: Thanks to authors. All comments have been addressed.

The manuscript bring up an interesting study related to the factors associated with breast lesions among women attending select teaching and referral health facilities in Kenya.

Reviewer #4: It is illogical to combine etiologic risk factors research for benign and malignant lesions, just because the lesions occur within the same organ. This explains the contrarian results obtained by the researchers.

Most breast lesions present in specific age periods, e.g. fibroadenomas - the commonest breast lump occurs in early adulthood while fibroadenosis occurs perimenopausally. While the latter is associated with nulliparity and low parity, the former is not. The association of breast cancer with age, parity, breast feeding and menopausal status are well known.

Reviewer #5: This manuscript investigates factors associated with breast lesions among women attending referral facilities in Kenya. It contributes valuable data from a low-resource setting and addresses an understudied area, particularly in Sub-Saharan Africa. The study design is appropriate, the statistical methodology is generally sound, and the authors have made significant improvements across three rounds of revision.

The authors have adequately addressed the previous comments, including clarifying the sample size justification, improving methodological transparency, enhancing the discussion section, and revising the language and structure of the manuscript. Although certain limitations remain (e.g., binary categorization of variables and lack of receptor status data), the authors have provided reasonable justifications for these constraints.

I recommend acceptance after minor revisions, primarily to enhance clarity on limitations and ensure consistency in presentation.

My comments for the author:

1 .Please consider briefly reiterating the limitations related to binary classification of variables (e.g., smoking, alcohol use) in the discussion section to enhance transparency.

2. Please clarify whether the classification of breast lesions was based on cytology, histopathology, or imaging, and whether standardized diagnostic criteria were applied consistently across study sites. The potential for interobserver variability or misclassification bias should be addressed.

3. The study assesses parity and contraceptive use as binary exposures, but does not account for timing or duration. For example, recent childbirth or long-term contraceptive use may have different biological impacts than remote exposures. Please discuss how this limitation might influence the findings.

Reviewer #6: This study looks at what causes breast lumps in Kenyan women by analyzing data from a cross-section of patients. It reveals some interesting connections between breast lumps and factors like how many children women have had, their exercise habits, and birth control use. What makes this research valuable is that it explores these health issues in a region where there hasn't been much research, helping us better understand what puts women at risk for both cancerous and non-cancerous breast problems.

Paper Weaknesses

1. Regarding reviewer 1, the author has not provided a convincing response. I recommend that the author increase the sample size, expand the analytical dimensions, update the statistical methods, and present the data more effectively.

2. The paper lacks detailed implementation guidelines, making it difficult for others to replicate the results. Providing additional details on hyperparameters, datasets, and code availability would enhance transparency.

3. How might the inclusion of receptor status data enhance the findings? Are there plans to incorporate this in subsequent research?

The paper should be submitted to another magazine with a lower impact factor.

**Do you want your identity to be public for this peer review?** For information about this choice, including consent withdrawal, please see our Privacy Policy

Reviewer #3: No

Reviewer #4: No

Reviewer #5: No

Reviewer #6: No

---

## [Author Response · Author response to Decision Letter 3]

25 Apr 2025

Dr. Josephine Nyabeta Rioki

Faculty of Health Sciences

Department of Human Pathology

University of Nairobi,

Kenya

25 April 2025.

Email: jrioki@uonbi.ac.ke

Dear reviewers,

PONE-D-24-31297R2: Factors associated with breast lesions among women attending select teaching and referral health facilities in Kenya: A cross-sectional study

We thank you for taking time to review our manuscript in consideration for publication in the PLoS ONE Journal. We are grateful for your observations and suggestions. Your comments were addressed as follows:

COMMENTS BY REVIEWER 3

Thanks to authors. All comments have been addressed.

The manuscript bring up an interesting study related to the factors associated with breast lesions among women attending select teaching and referral health facilities in Kenya.

Response:

We appreciate the reveiwer’s sentiments.

COMMENTS BY REVIEWER 4

It is illogical to combine etiologic risk factors research for benign and malignant lesions, just because the lesions occur within the same organ. This explains the contrarian results obtained by the researchers.

Most breast lesions present in specific age periods, e.g. fibroadenomas - the commonest breast lump occurs in early adulthood while fibroadenosis occurs perimenopausally. While the latter is associated with nulliparity and low parity, the former is not. The association of breast cancer with age, parity, breast feeding and menopausal status are well known.

Response:

Actually, in concurrence with the reviewer’s assertion that the factors studied may have distinct effects depending on the lesion(s) under investigation, a multinomial logistic regression framework was employed since it permits the estimation of separate effects (odds ratios) of risk factors for each class of lesions. Moreover, considering the cross-sectional nature of the study and that study participants were cytologically classified into one of four possible lesion categories, it was deemed statistically sensible to assess the factor-lesion associations in a combined multinomial fashion. Furthermore, even though a number of these factors have been studied in other population settings, there is scarcity of evidence of factors associated with breast lesions amongst women in Sub-Saharan Africa.

COMMENTS BY REVIEWER 5

This manuscript investigates factors associated with breast lesions among women attending referral facilities in Kenya. It contributes valuable data from a low-resource setting and addresses an understudied area, particularly in Sub-Saharan Africa. The study design is appropriate, the statistical methodology is generally sound, and the authors have made significant improvements across three rounds of revision.

The authors have adequately addressed the previous comments, including clarifying the sample size justification, improving methodological transparency, enhancing the discussion section, and revising the language and structure of the manuscript. Although certain limitations remain (e.g., binary categorization of variables and lack of receptor status data), the authors have provided reasonable justifications for these constraints.

I recommend acceptance after minor revisions, primarily to enhance clarity on limitations and ensure consistency in presentation.

My comments for the author:

1 .Please consider briefly reiterating the limitations related to binary classification of variables (e.g., smoking, alcohol use) in the discussion section to enhance transparency.

2. Please clarify whether the classification of breast lesions was based on cytology, histopathology, or imaging, and whether standardized diagnostic criteria were applied consistently across study sites. The potential for interobserver variability or misclassification bias should be addressed.

3. The study assesses parity and contraceptive use as binary exposures, but does not account for timing or duration. For example, recent childbirth or long-term contraceptive use may have different biological impacts than remote exposures. Please discuss how this limitation might influence the findings.

Response:

We are appreciative of the positive feedback given by the reviewer. On point 1, the mentioned limitations have been added in the discussion segment – see lines 322-334. On point 2, the outcome definition has been refined and further information provided – see lines 126-129. On point 3, the impacts of recency of child birth and long-term contraceptive use have been explained – see lines 235-237 & 259-260.

COMMENTS BY REVIEWER 6

This study looks at what causes breast lumps in Kenyan women by analyzing data from a cross-section of patients. It reveals some interesting connections between breast lumps and factors like how many children women have had, their exercise habits, and birth control use. What makes this research valuable is that it explores these health issues in a region where there hasn't been much research, helping us better understand what puts women at risk for both cancerous and non-cancerous breast problems.

Paper Weaknesses

1. Regarding reviewer 1, the author has not provided a convincing response. I recommend that the author increase the sample size, expand the analytical dimensions, update the statistical methods, and present the data more effectively.

2. The paper lacks detailed implementation guidelines, making it difficult for others to replicate the results. Providing additional details on hyperparameters, datasets, and code availability would enhance transparency.

3. How might the inclusion of receptor status data enhance the findings? Are there plans to incorporate this in subsequent research?

The paper should be submitted to another magazine with a lower impact factor.

Response:

Point 1: As noted in our previous rebuttal, on sample size, Schwab (2002) contends that for a multinomial logistic regression, a minimum of 10 cases per independent variable is necessary to sufficiently power it. With 12 predictors in our analysis, this translates to a sample of 120. Thus, our sample of 651 individuals adequately fulfils this requirement. On ‘analytical dimensions’, ‘update the statistical methods’, and ‘present the data more effectively’, it is not clear what the reviewer specifically seeks to be addressed. Surprisingly, though, on the ‘comments to the author’ (point 3), the reviewer agrees that ‘the statistical analysis has been performed appropriately and rigorously’. Nevertheless, it is worth noting that we have endeavoured to strictly adhere to established guidelines for the reporting of cross-sectional studies to allow reproducibility of the study findings – see line 107.

Point 2: On ‘implementation guidelines’, it is unclear what the reviewer means here in reference to the study. Nonetheless, as aforementioned, the study has strictly complied to STROBE guidelines for reporting epidemiological studies to permit replicability of the results. On ‘hyperparameters’, these relate to the field of Bayesian statistics. It is important to note that our study applied a frequentist approach. It is apparent to the authors that some of the reviewer’s comments may not relate to the present study. On ‘datasets’, ‘code availability’, these had previously been deposited in a public data repository (Harvard Dataverse: https://doi.org/10.7910/DVN/2OXWTA) – but we have also included this information under the statistical section of the manuscript – see line 149. It is also worthwhile to note that on the ‘comments to the author’ (point 4), the reviewer agrees that ‘all data underlying the findings in the manuscript’ have been made available.

Point 3: On receptor status, this limitation has now been pointed out in the manuscript – see lines 328-334.

On whether the manuscript should be submitted to ‘another magazine’, we remain persuaded that our study conforms to the scope of PLOS One journal.

Reference:

Schwab, J.A. 2002. Multinomial Logistic Regression: Basic Relationships and Complete Problems. University of Texas, Austin, Texas

Again, we wish to thank the reviewers for their thoughtful feedback.

Yours faithfully,

Dr. Josephine N. Rioki

---

## [Editor Report · Decision Letter 3]

13 May 2025

Factors associated with breast lesions among women attending select teaching and referral health facilities in Kenya: A cross-sectional study

PONE-D-24-31297R3

Dear Dr. Rioki,

We’re pleased to inform you that your manuscript has been judged scientifically suitable for publication and will be formally accepted for publication once it meets all outstanding technical requirements.

Kind regards,

Jie Yang, M.D.

Guest Editor

PLOS ONE

Additional Editor Comments (optional):

Thanks for the authors' efforts to comprehensively improve your manuscript according to editor's and reviewers' comments. I am pleased to inform you that your paper can be accepted for publication now. Thanks for the chance to assess your important work. Additionally, many thanks for all the reviewers' precious inputs.
---

## [Editor Report · Acceptance letter]

PONE-D-24-31297R3

PLOS ONE

Dear Dr. Rioki,

I'm pleased to inform you that your manuscript has been deemed suitable for publication in PLOS ONE. Congratulations! Your manuscript is now being handed over to our production team.

Kind regards,

on behalf of

Dr. Jie Yang

Guest Editor

PLOS ONE